# OpenReview forum: "“Kelly is a Warm Person, Joseph is a Role Model”: Gender Biases in LLM-Generated Reference Letters"
_EMNLP/2023/Conference — EMNLP 2023 Findings_

### Official Review · Reviewer_rSCW · 2023-07-30

**Soundness:** 3

**Excitement:**

3: Ambivalent: It has merits (e.g., it reports state-of-the-art results, the idea is nice), but there are key weaknesses (e.g., it describes incremental work), and it can significantly benefit from another round of revision. However, I won't object to accepting it if my co-reviewers champion it.

**Paper Topic And Main Contributions:**

This paper presented an analysis of gender biases in generative languages models such as ChatGPT and Alpaca when prompted for the generation of recommendation letters. The authors presented an analysis of gender biases at lexical and stylistic levels. In particular, it was interesting to see that showed models such as ChatGPT, Alpca, and other recent models, tend to produce text with a more formal style and positive sentiment when referring to a male person compared to a female person.

**Questions For The Authors:**

In Table 1, the generated letter for Kelly doesn’t use the word “warm” but the caption mentions its usage.

In line 335, the notation denotes D_1, D_2, .. D_n as sentences  while D represents a document, which is a confusing notation. I would suggest using different lettering for sentences/documents.

The citation “Transformer et al.” on line 37 be fixed. There are also similar issues with other references where year doesn’t appear or the publication venue is missing in the list of references.


**Reasons To Accept:**

The paper highlights the important issue of potentially biased LLM-generated recommendation letters. Such a finding is important, since there seems to be increased use of LLMs for this purpose, such gender biases could lower success rates for female applicants. I believe the study presented in the paper could stir more research on those issues LLMs when used as writing assistants.

**Reasons To Reject:**

The following are some concerns about the paper:

While gender biases indeed exist in the generations produced by language models when prompted to write text that refer to a male or female person and should be mitigated, one particular concern is about the ethicality of using LLMs for automatically writing recommendation letters, which in my personal opinion I find unethical. However, Despite that, the paper appears to promote LLM-generated recommendation letters whereas I would recommend reframing the paper as to point out a critical issue in an area where ethics are of concern.


Sentiment analysis is considered among style-related analysis, however sentiment should be decoupled from style since it holds a more functional than aesthetic role in text. My suggestion would be to focus on more stylistic elements besides formality which could reveal further interesting findings.


**Reproducibility:**

4: Could mostly reproduce the results, but there may be some variation because of sample variance or minor variations in their interpretation of the protocol or method.

**Reviewer Confidence:**

4: Quite sure. I tried to check the important points carefully. It's unlikely, though conceivable, that I missed something that should affect my ratings.

---

> ### Author Rebuttal · Authors · 2023-08-29
>
> We would like to first thank the reviewer for the constructive and valuable feedback. Below, we address the concerns and questions raised by the reviewer.
>
> * Regarding your concern on the **ethical issues of using LLM to write recommendation letters**, we fully agree with your point that ethical issues are concerning in the use case of LLM-generated recommendation letters, and would like to clarify that the purpose of our work is not to promote LLM-generated letters, but to point out potential biases, harms and dangers of such applications in the real world without careful investigation. In the introduction, we stated that ‘current LLMs are not yet adequately mature for generating professional documents’ (lines 92-94), and that we ‘highlight the risk of potential harm when they are employed in such real-world applications’ (lines 94-96). The motivation behind running experiments and evaluating biases in generated recommendation letters in this study is also to reveal the dangers, biases and ethical concerns in this real-world application of LLMs. We will make modifications to further improve our writing and clarify our main point that gender biases in LLM-generated documents is a critical issue that is in urgent need to be thoroughly studied and employed judiciously. Nevertheless, we are delighted to receive your recognition on the research direction, and are confident that our work will pave the way for future researchers that further studies methods to evaluate and mitigate gender biases in LLM-generated professional documents.
>
> * Regarding your concerns on **style-related analysis**, we utilize language sentiment as a proxy to reflect the level of excellency in language (lines 296-298), or how ‘excellent’ and capable is the language used to describe the candidate in the letters. We believe that this aligns with the other two components of the language style analysis: language formality, which acts as a proxy in our study to reflect level of language professionalism (lines 286-288), or how professional is the language used in the generated letters; language agency, which describes whether the language used in the letter is leaning towards being ‘agentic’ or ‘communal’. These writing styles have been noted to be biased and cause direct allocational harm to individuals from social science literature (Dutt et al. 2016., Liu et al. 2019., Khan et al. 2021., Trix et al. 2003., Madera et al. 2009., Madera et al. 2019). In order to better convey this definition to readers, we will adjust our writing to further clarify these aspects of analysis in the revised version of our paper. Also, we will include additional stylistic elements such as document and sentence length across generated reference letters.
>
> * Regarding your question on the **description for 'Kelly'**, we use ‘warm’ to describe the general personality trait of Kelly in the generated letter (e.g. excellent interpersonal skills, a well-liked member). We don't mean that the word "warm" appears exactly in the generated letter. We will clarify this in the revision or change the title to avoid confusion.
>
> * Regarding your advice on **notation**, we appreciate the suggestion and will change our notation of sentences in document D into $s_1$, $s_2$, …. To avoid confusion with the scoring function, we will use Score($\cdot$) as the new notation of the scoring function.
>
> * Last but not least, we appreciate your detailed point on issues with **references**, and will fix them in the revised version.
>
> References:
>
> [1] Dutt, K., Pfaff, D., Bernstein, A. et al. Gender differences in recommendation letters for postdoctoral fellowships in geoscience. Nature Geosci 9, 805–808 (2016). https://doi.org/10.1038/ngeo2819
>
> [2] Liu, O. L., Minsky, J., Ling, G., & Kyllonen, P. (2009). Using the Standardized Letters of Recommendation in selection: Results from a multidimensional Rasch model. Educational and Psychological Measurement, 69(3), 475–492. https://doi.org/10.1177/0013164408322031)
>
> [3] Shawn Khan, Abirami Kirubarajan, Tahmina Shamsheri, Adam Clayton, and Geeta Mehta. 2021. Gender bias in reference letters for residency and academic medicine: a systematic review. Postgraduate Medical Journal.
>
> [4] Frances Trix and Carolyn E. Psenka. 2003. Exploring the color of glass: Letters of recommendation for female and male medical faculty. Discourse & Society, 14:191 – 220.
>
> [5] Juan Madera, Mikki Hebl, and Randi Martin. 2009. Gender and letters of recommendation for academia: Agentic and communal differences. The Journal of applied psychology, 94:1591–9.
>
> [6] Juan Madera, Mikki Hebl, Heather Dial, Randi Martin, and Virginia Valian. 2019. Raising doubt in letters of recommendation for academia: Gender differences and their impact. Journal of Business and Psychology, 34.

---

### Official Review · Reviewer_kHWV · 2023-08-01

**Typos Grammar Style And Presentation Improvements:** 1. **Question on this task and settin…
**Soundness:** 4

**Excitement:**

3: Ambivalent: It has merits (e.g., it reports state-of-the-art results, the idea is nice), but there are key weaknesses (e.g., it describes incremental work), and it can significantly benefit from another round of revision. However, I won't object to accepting it if my co-reviewers champion it.

**Paper Topic And Main Contributions:**

This paper mainly addresses the gender bias phenomenon in LLM-generated reference letters. The author designed intuitive evaluation methods: namely context-based/context-less generation,  and proposed some structured multi-dimensional metrics (lexical/style/formality) to comprehensively reveal the bias problem of current widely-adopted LLMs (e.g., ChatGPT and Alpaca), including bias propagation and amplification. The author also collected a corpus via some automated strategies for this specific problem.

**Reasons To Accept:**

1. **An interesting gender bias scenario for LLMs**. To my knowledge, this might be the first work to research the bias in *reference letter generation*.
2. **Reasonable experiment scheme**. This paper proposes a comprehensive evaluation framework; some metrics seem to be novel (such as the "language agency" might be a brand-new axis for this area).

**Reasons To Reject:**

1. **Limited contribution**. Though this paper devises a comprehensive scheme for evaluating gender bias in LLM-generated reference letters, the overall contribution made by this work is still limited:
- Most methods used in this work are widely adopted strategies, such as odd ratios.
- This paper only *identifies* this problem, but there are no other insights or inspirations concluded to help *address* this problem. For example, how to design novel prompting strategies (e.g., iterative prompt, explicit rules) to significantly reduce the bias ratios.

2. **Some writing improvement**. Overall, someone would find it hard to follow this paper. Pls refer to my detailed comments in *"Typos Grammar Style And Presentation Improvements"* section.

3. **Concern about the data quality and base model performance.** In this paper, the author used some automated pipelines to construct the dataset (such as using Spacy to extract and replace entities). However, I conjecture that there should be a lot of noise in this kind of dataset, which could affect the experiment analysis. Besides, some base models can also introduce some unexpected bias; for example, the agentic classifier only trained on 600 ChatGPT-synthetic data, and the author didn't report the test or eval performances.



**Reproducibility:**

3: Could reproduce the results with some difficulty. The settings of parameters are underspecified or subjectively determined; the training/evaluation data are not widely available.

**Reviewer Confidence:**

3: Pretty sure, but there's a chance I missed something. Although I have a good feel for this area in general, I did not carefully check the paper's details, e.g., the math, experimental design, or novelty.

---

> ### Author Rebuttal · Authors · 2023-08-29
>
> We would like to first thank the reviewer for the constructive and valuable feedback. Below, we address the concerns and questions raised by the reviewer.
>
> * Regarding concerns on **contributions**, we believe that the major contribution of our work is being among the first to point out biases and harms with LLM-generated recommendation letters, as well as construct a comprehensive framework for evaluating different aspects of biases. Grounding our insights on social science literature, we are the first to consolidate the task setting and propose well-defined metrics (Biases in Lexical Content, which includes analysis of nouns and adjectives; Biases in Language Style, which includes analysis of biases in language formality, language sentiment, as well as language agency) to evaluate gender biases in LLM-generated letters. Furthermore, we are the first to propose the study of hallucination biases in LLMs, and revealed through experiments that LLMs are significantly amplifying gender biases in the hallucinated contents of their generations. In addition to using widely adopted strategies such as Odds Ratio, this work further proposes a new metric to evaluate biases in language style $b_{lang}$, as discussed in section 3.2.2.
>
> * As for the concern on **insights and inspirations to help address this problem**, we will include further discussions of potential directions to mitigate this bias through prompting strategies such as iterative prompt and explicit rules. We believe that our work provides a systematic framework for evaluation of gender biases in LLM-generated recommendation letters, and that our contribution will raise awareness of this imminent issue from the NLP fairness community, leading to future research works that further studies holistic mitigation methods for this problem.
>
> * Regarding your concern about **data quality and base model**, we will make efforts to: i) provide full details of performances on both validation and test datasets for all three classifiers in the revised version of our paper, ii) release the generated dataset as well as the checkpoint of the trained language agency classifier upon acceptance, to ensure reproducibility of our experiment results, and iii) address potential noise and unexpected biases of base models in the Limitations section. We will also clarify and emphasize the settings of parameters and approaches for choosing them from Appendix F.2 (lines 1122-1129)
>
> * Regarding your question on **task and settings**, we acknowledge that both CLG and CBG can be adopted in real-world use cases. With CBG, we wanted to simulate a user scenario where they would write a short description of themselves and ask the model to generate the recommendation letter. We acknowledge that this may not always be the case due to the nature of prompting and usage of these models. In both settings, however, our experiment results reveal significant gender biases, which further highlights this imminent problem in LLM applications. In our revised version, we will adjust our writing to clarify that both CLG and CBG can be related to real-world usage scenarios, and that our experiments reveal significant gender biases in both task settings.
>
> * Regarding your comments on **choice of LLMs**, we appreciate your valuable insights on using dialogue-like prompts for Vicuna. In our work, we didn’t choose to further experiment with Vicuna since we hope to establish a fair comparison through evaluating LLMs in the same settings (using instruction to prompt the generation of recommendation letters), under which Vicuna suffers from a low Generation Success Rate (as shown in Appendix B.3 and Table 7). In the revised version of our paper, we are happy to add discussions on the potential alternative method of using dialogue-like prompts for Vicuna, as well as presenting preliminary performance results using such prompts. We will also add discussions on the potential of using novel prompting methods to address bias problems.
>
> * Regarding your advice on writing, we appreciate the detailed suggestion and will make modifications accordingly in the revised version of our paper to improve readability and flow.

---

### Official Review · Reviewer_yJSC · 2023-08-04

**Soundness:** 4

**Excitement:**

4: Strong: This paper deepens the understanding of some phenomenon or lowers the barriers to an existing research direction.

**Paper Topic And Main Contributions:**

This paper aims to examine gender bias in Large Language Model (LLM)-generated reference letters in two different prompting settings: (1) context-less (minimal context given) generation and (2) context-based (guided by hobbies...) generation. The two examined dimensions are biases in: (1) language styles (formality, sentiment, and agency classifiers) and (2) lexical content (frequency differences between nouns/adjectives in male and female documents). The authors also measure the amplification of bias hallucinated content. The results of this paper indicate that when given insufficient context, LLMs default to generating content based on gender stereotypes. Even when detailed information about the subject is provided, they tend to employ different word choices and linguistic styles when describing candidates of different genders. Additionally, biases are amplified when models hallucinate.

**Questions For The Authors:**

(1) Dataset construction methodology: How was the agency classification corpus collected, annotated, and preprocessed to ensure reliability and quality?

(2) Inter-rater agreement: How was the inter-rater agreement assessed for the labeled datasets used in this study?

(2) Classifier performance: What are the precision, recall, and F1 scores metrics for the formality, sentiment, and agency classifiers?

(3) Language style definition: How do you define and categorize formality, sentiment, and agency in the context of this study?

**Reasons To Accept:**

The main strengths of this work pertain to the novelty of the bias evaluation technique (language style and lexical content gaps between genders) as well as the correlation found between hallucination and amplified bias.

**Reasons To Reject:**

The only weakness of this work pertains to the over-reliance on off-the-shelf classifiers/datasets to compute formality, sentiment, and agency as well as the commission of critical information of the models used to measure language styles. The authors use Grammarly’s Yahoo Answers Formality Corpus (GYAFC) to train a formality classifier, SST-2 dataset to train a sentiment classifier, and use ChatGPT to synthetically generate an agency classification corpus. The paper fails to report any information on the dataset construction methodology, inter-rater agreement, performance, and error analysis of the trained classifiers. Given that the scores outputted from the three models heavily influence the resultant bias scores, the authors aught to provide more information and evidence relating to their efficacy as well as a clear definition and categorization of each linguistic phenomenon.

**Reproducibility:**

2: Would be hard pressed to reproduce the results. The contribution depends on data that are simply not available outside the author's institution or consortium; not enough details are provided.

**Reviewer Confidence:**

4: Quite sure. I tried to check the important points carefully. It's unlikely, though conceivable, that I missed something that should affect my ratings.

---

> ### Author Rebuttal · Authors · 2023-08-29
>
> We would like to first thank the reviewer for the constructive and valuable feedback. Below, we address the concerns and questions raised by the reviewer.
>
> * Regarding your concern on the **dataset construction methodology**, we have discussed the collection process of the agency classification dataset in Appendix F.2: we synthetically generated the agency classification dataset through prompting ChatGPT to rephrase an initial biography from the Bias in Bios dataset into i) one version in agentic language style, and ii) a second version in communal language style. To improve the reliability, consistency, and quality of generation, we made sure to include clear definitions of agentic and communal language from social science literature in the prompt. We will make sure to further clarify the collection process by providing the following full prompt used for the dataset synthesis process in the appendix section:
>
>       Prompt: “You will rephrase a biography two times to demonstrate agentic and communal language traits respectively. "agentic" is defined as more achievement-oriented, and "communal" is defined as more social or service-oriented. Output your answer in a json format: {"agentic": {agentic bio}, "communal": {communal bio}}. The biography is:”
>
> * Regarding your question on the **inter-rater agreement** on the datasets used, we would like to clarify that we did not construct the GYAFC dataset or the SST-2 dataset, but followed previous works (Rao & Tetreault, 2018., Socher et al. 2013.) to use their constructions of the two datasets. For the language agency classification dataset, we used ChatGPT to generate the agentic and communal version of biographies. To ensure reliability, consistency, and quality of generation, we condition ChatGPT’s outputs on specific definitions of agentic and communal language in social science literature, as mentioned in the previous paragraph. We additionally invited 2 expert annotators to conduct human evaluation of a held-out test set of 61 samples (around 10% of our 600 generations) from the generated dataset. Specifically, each expert is asked to manually label the test set. The mean expert-dataset agreement score using Cohen’s Kappa is 0.864 (one 0.829 and one 0.899) and the inter-researcher agreement score using Cohen’s Kappa between the two experts is 0.862. Fleiss’s Kappa agreement score between the two expert annotators and the dataset labels is 0.863. All agreement scores demonstrate good levels of inter-rater and rater-dataset alignment, proving the satisfactory quality of the synthesized agency classification dataset. We will add an additional report and details of the human evaluation process in the appendix of the revised version of our paper, in order to emphasize the quality and trustworthiness of the synthesized dataset. We will also release the generated dataset upon acceptance, as mentioned in lines 99-100, to ensure reproducibility.
>
> * Regarding your question on trained classifier statistics, we hereby provide full details of the precision, recall and F1 score metrics for all three classifiers. ‘Language Agency’ dataset refers to the language agency classification dataset that we synthesized in this work:
>
> | Classifier  | Dataset | Precision | Recall | F1 | Accuracy
> | :--------: | :-------: | :-------: | :-------: | :-------: | :-------: |
>  Formality  | GYAFC  | 0.90  | 0.91 | 0.90 | 0.91
>  Sentiment   | SST-2 | 0.99 | 0.99 | 0.99 | 0.99
>  Agency | Language Agency | 0.92 | 1.00 | 0.96  | 0.96
>
> We will include the details of classifier statistics in the appendix section of the revised version of our paper. We will release the checkpoint of the trained language agency classifier upon acceptance, as mentioned in lines 99-100, to ensure reproducibility of our experiment results.
>
> * Regarding your question on **language style definition**, we provided definitions in section 2.2 and section 3.2.2: language formality in this study acts as a proxy to reflect level of language professionalism (lines 286-288), or how professional is the language used in the generated letters. Language sentiment in this study acts as a proxy to reflect the level of excellency in language (lines 296-298), or how ‘excellent’ and capable is the language used to describe the candidate in the letters. Language agency describes whether the language used in the letter is leaning towards being ‘agentic’ or ‘communal’. Agentic characteristics include speaking assertively, influencing others, and initiating tasks (lines 178-180), whereas communal characteristics include concerning with the welfare of others, helping others, accepting others’ direction, and maintaining relationships (lines 180-183). We will make sure to further clarify the definitions in the Methods section in the revised version, in order to better convey the concepts to readers. We also would like to point out that the aspects of these metrics are grounded in social science literature and are empirically proven to have biases as well as direct allocational harms in prior studies.
>
> References:
>
> [1] Sudha Rao and Joel Tetreault. 2018. Dear Sir or Madam, May I Introduce the GYAFC Dataset: Corpus, Benchmarks and Metrics for Formality Style Transfer. In Proceedings of the 2018 Conference of the North American Chapter of the Association for Computational Linguistics: Human Language Technologies, Volume 1 (Long Papers), pages 129–140, New Orleans, Louisiana. Association for Computational Linguistics.
>
> [2] Richard Socher, Alex Perelygin, Jean Wu, Jason Chuang, Christopher D. Manning, Andrew Ng, and Christopher Potts. 2013. Recursive Deep Models for Semantic Compositionality Over a Sentiment Treebank. In Proceedings of the 2013 Conference on Empirical Methods in Natural Language Processing, pages 1631–1642, Seattle, Washington, USA. Association for Computational Linguistics.

---

### Meta-Review · Area_Chair_awyX · 2023-09-18

**Recommendation:** 4

**Metareview:**

Overall, the reviewers found this work to be sound, the proposed methodology to be novel, and the findings to be interesting. The impact of the paper also has practical/societal significance and may help to promote even more work in the area in the future. Some of the reviewers were not quite as excited because of the perceived limited methodological contribution (i.e., no new methods were introduced and existing ones were used), though I don’t see this as a pure methods paper so I don’t believe that should be heavily weighted. On the other hand, a reviewer points out that there are limited insights into how to address the problem that is pointed out. The authors make a note that they will address this further if the paper is accepted. Other comments regarding the framing of the paper (appears to promote using LLMs for reference letters, referring to sentiment as a kind of “style”) are valid concerns but also appear to be addressed sufficiently by the authors in their rebuttal. The paper may benefit from some additional writing improvements as recommended by the authors, but these appear to be mostly minor (yet still important to address!).

---

### Decision · Program_Chairs · 2023-10-07

**Decision:**

Accept-Findings

**Comment:**

Overall, the reviewers found this work to be sound, the proposed methodology to be novel, and the findings to be interesting. The impact of the paper also has practical/societal significance and may help to promote even more work in the area in the future. Some of the reviewers were not quite as excited because of the perceived limited methodological contribution (i.e., no new methods were introduced and existing ones were used), though I don’t see this as a pure methods paper so I don’t believe that should be heavily weighted. On the other hand, a reviewer points out that there are limited insights into how to address the problem that is pointed out. The authors make a note that they will address this further if the paper is accepted. Other comments regarding the framing of the paper (appears to promote using LLMs for reference letters, referring to sentiment as a kind of “style”) are valid concerns but also appear to be addressed sufficiently by the authors in their rebuttal. The paper may benefit from some additional writing improvements as recommended by the authors, but these appear to be mostly minor (yet still important to address!).